# Significance of P53-Binding Protein 1 as a Novel Molecular Histological Marker for Hypopharyngeal Squamous Neoplasms

**DOI:** 10.3390/cancers16172987

**Published:** 2024-08-28

**Authors:** Hiroko Kawasaki-Inomata, Maiko Tabuchi, Kiyuu Norimatsu, Tetsuro Honda, Katsuya Matsuda, Keiichi Hashiguchi, Naoyuki Yamaguchi, Hideaki Nishi, Yoshihiko Kumai, Masahiro Nakashima, Hisamitsu Miyaaki, Kazuhiko Nakao, Yuko Akazawa

**Affiliations:** 1Department of Gastroenterology and Hepatology, Nagasaki University Graduate School of Biomedical Sciences, Nagasaki 852-8501, Japan; hkawasaki@nagasaki-u.ac.jp (H.K.-I.); m.tabuchi@nagasaki-u.ac.jp (M.T.); khashiguchi@nagasaki-u.ac.jp (K.H.); naoyuki3435@nagasaki-u.ac.jp (N.Y.); miyaaki-hi@nagasaki-u.ac.jp (H.M.); kazuhiko@nagasaki-u.ac.jp (K.N.); 2Department of Histology and Cell Biology, Nagasaki University Graduate School of Biomedical Sciences, Nagasaki 852-8523, Japan; 3Department of Rheumatology, National Hospital Organization Ureshino Medical Center, Saga 843-0393, Japan; norimatsu.kiyu.aj@mail.hosp.go.jp; 4Medical Education Development Center, Nagasaki University Hospital, Nagasaki 852-8501, Japan; htetsu0319@gmail.com; 5Department of Tumor and Diagnostic Pathology, Atomic Bomb Disease Institute, Nagasaki University, Nagasaki 852-8523, Japan; katsuya@nagasaki-u.ac.jp (K.M.); moemoe@nagasaki-u.ac.jp (M.N.); 6Department of Otolaryngology, Head and Neck Surgery, Nagasaki University Graduate School of Biomedical Sciences, Nagasaki 852-8501, Japan; nishi2424@nagasaki-u.ac.jp (H.N.); ykumai426@nagasaki-u.ac.jp (Y.K.); 7Department of Gastroenterology, Sasebo City General Hospital, Nagasaki 857-8511, Japan

**Keywords:** hypopharyngeal squamous cell carcinoma, p53-binding protein 1, DNA double-strand break, genomic instability

## Abstract

**Simple Summary:**

The DNA damage response protein p53-binding protein 1 (53BP1) exhibits abnormal foci in the nucleus during carcinogenesis in various organs. However, the pathophysiological changes that occur during the carcinogenic process of hypopharyngeal squamous cell carcinoma remain unclear. This study revealed a stepwise increase in aberrant 53BP1 expression on the tumor surface during carcinogenesis. In addition, a significant difference in the co-expression of 53BP1 and Ki67 was observed on the tumor surface when the tumor thickness exceeded 1000 µm, which is considered the threshold at which the risk of lymph node metastasis and vascular invasion increases in hypopharyngeal cancer. These findings suggest that 53BP1 could be useful for the pathological diagnosis of hypopharyngeal cancer and for predicting the prognosis of patients with this disease.

**Abstract:**

The DNA damage response protein p53-binding protein 1 (53BP1) accumulates and forms foci at double-strand DNA breaks, indicating the extent of DNA instability. However, the potential role of 53BP1 as a molecular biomarker for hypopharyngeal squamous cell carcinoma (HPSCC) diagnosis remains unknown. Here, we evaluated the potential of immunofluorescence-based analysis of 53BP1 expression to differentiate the histology of hypopharyngeal neoplasms. A total of 125 lesions from 39 surgically or endoscopically resected specimens from patients with HPSCC was histologically evaluated. 53BP1 expression in the nucleus was examined using immunofluorescence. The number of 53BP1 nuclear foci increased with the progression from non-tumorous to low-grade dysplasia, high-grade dysplasia, and squamous cell carcinoma. Unstable 53BP1 expression served as an independent factor for distinguishing lesions that required intervention. Colocalization of 53BP1 foci in proliferating cells, as assessed by Ki67, was increased in tumors ≥ 1000 µm in depth compared to those <1000 µm in depth at the tumor surface. Hence, the expression patterns of nuclear 53BP1 foci were associated with the progression of hypopharyngeal neoplasms. These findings suggest that 53BP1 could serve as an ancillary marker to support histological diagnosis and predict the factors that influence prognosis in patients with HPSCC.

## 1. Introduction

Hypopharyngeal squamous cell carcinoma (HPSCC) is often diagnosed at advanced stages and has one of the poorest prognoses among head and neck cancers [1,2,3,4]. However, recent improvements in endoscopic diagnostic procedures have enabled the detection of HPSCC during esophagogastroduodenoscopy (EGD) [5,6]. The disease originates from epithelial dysplasia attributed to the accumulation of genetic changes that predispose patients to a higher risk of progression from low-grade dysplasia (LD) to high-grade dysplasia (HD) [7,8].

Difficulties in endoscopic procedures and limited hypopharyngeal biopsy sample sizes often result in inconsistent histological dysplasia grading [9,10]. The lack of reliable diagnostic markers for grading precancerous lesions further limits such analyses [8]. While guidelines for the endoscopic treatment of pharyngeal cancer have not been established, previous epidemiological follow-up studies on esophageal cancer have indicated that HD in the squamous epithelium serves as a notable indication for therapeutic intervention owing to its potential to progress to malignancy as well as squamous cell carcinoma (SCC) in the esophagus [11,12]. In contrast, LD has a lower potential for progression to carcinoma [11,13]. In Japan, the endoscopic diagnosis of pharyngeal lesions tends to follow a protocol similar to that for esophageal SCC. Moreover, minimally invasive endoscopic treatments, such as endoscopic submucosal dissection (ESD) or endoscopic laryngopharyngeal surgery (ELPS), are increasingly being utilized for HPSCC without lymph node metastasis in pharyngeal neoplasm. HPSCC tumors with a thickness of ≥1000 µm are at a greater risk of lymph node metastasis and vascular invasion [14,15,16]. However, no standardized protocols that precisely predict the malignant potential and tumor depth or vascular invasion are currently available. Therefore, the development of additional histological strategies for the accurate diagnosis and prognosis prediction of HPSCC is of significant clinical interest.

An impaired DNA damage response results in genomic instability and is widely regarded as a crucial event in carcinogenesis, including HPSCC development [17,18]. Alcohol and tobacco induce DNA damage and instability during HPSCC carcinogenesis [19,20]. Additionally, DNA instability may result in dysfunctional repair mechanisms, which may cause further the aggregation of the damaged DNA [21].

Tumor protein 53-binding protein 1 (53BP1) is a nuclear protein that accumulates and forms foci in response to double-strand DNA breaks along with other DNA damage response molecules, including gamma histone 2AX (γH2AX) and ataxia telangiectasia mutated (ATM) [22,23,24]. Abnormal findings in immunohistochemical analyses of 53BP1 foci, including an increase in the number and size, are putative molecular markers of DNA damage response during carcinogenesis. We reported an increase in such abnormalities during the progression of squamous neoplasms of the esophagus [25], uterus [26], oral cavity [9], and thyroid gland [27]. These findings correlated well with chromosome copy number aberrations, indicating that abnormal 53BP1 foci may serve as molecular markers of genomic instability during carcinogenesis. Given the currently unsubstantiated potential utility of 53BP1 as a molecular biomarker for the diagnosis and prognosis of HPSCC, we aimed to analyze 53BP1 expression patterns during carcinogenesis and its association with clinicopathological features in affected individuals.

## 2. Materials and Methods

### 2.1. Patients

This retrospective study included patients who underwent ESDs or surgical resections at Nagasaki University Hospital between 2017 and 2019. This study was approved by the Institutional Ethical Committees for Medical Research at Nagasaki University Hospital (21101916-2) and the Graduate School of Biomedical Sciences, Nagasaki University (15062617-4). We were granted a waiver of participant consent for the following reasons: the study was anonymized, no personal identifiers were collected, no additional specimens were obtained from participants, and the study was retrospective. The study was conducted in accordance with the Declaration of Helsinki and adhered to the ethical committee’s official informed consent and disclosure system guidelines. The study information was accessible on the institution’s website, and patients were allowed to withdraw consent to participate. Patients younger than 20 years and histological specimens with severe burns caused by the procedure were excluded from the study. Hence, a total of 125 lesions obtained via ESD from 24 patients or surgical intervention from 15 patients, encompassing 39 patients with HPSCC, was included. Clinical data, including age, sex, smoking history, alcohol consumption, and the presence of other malignancies, were recorded during routine clinical analysis.

### 2.2. Histological Evaluation

Histological diagnosis and classification of resected specimens were performed according to the two-tiered system of the World Health Organization (WHO) 2017 classification of head and neck tumors [7]. Precursor lesions were classified as either LD or HD. LD was diagnosed based on lesions exhibiting minimal atypia up to the lower half of the epithelium, and retention of maturation in the upper half. HD was diagnosed based on lesions with cellular and nuclear atypia involved in ≥ half of the epithelial thickness without stromal alterations. SCC was diagnosed when prominent architectural and cellular abnormalities of the entire epithelium were evident with pronounced nuclear and cellular atypia. Representative images are shown in Figure 1. In this study, all cases were stained with D2-40 (lymphatic) and Elastica van Gieson (EVG, blood vessels), in addition to routine Hematoxylin and Eosin (HE) staining, allowing for differentiation between lymphatic and vascular invasion. Invasion of the vessels was only confirmed when tumor cells were within the vascular space. Tumor thickness was defined as the distance from the tumor surface to its deepest point according to the General Rules for Clinical Studies on Head and Neck Cancer, 6th Edition, Revised Version (Japan Society for Head and Neck Cancer, December 2019, 86) [14,15,16,28]. The diagnosis was made by two independent pathologists, Y.A. and M.N. The number of nuclei within each region was quantified, and the percentage of nuclei that were expressed by immunofluorescence staining of 53BP1 and/or Ki67 was calculated. The most representative areas for each histology were randomly selected as described previously [9,25,26,27,29,30]. Lesions were classified as non-neoplastic from outside the tumor margin (38 sites/38 cases), LD (14 sites/14 cases), HD (8 sites/8 cases), or SCC (surface lesion: 30 sites/30 cases, invasive front: 35 sites/35 cases), with a total of 125 sites (Appendix A).

### 2.3. Immunofluorescence Analysis

Double-labeled immunofluorescence staining of 53BP/Ki67 was performed. Deparaffinized tissue sections were preincubated with 10% normal goat serum after antigen retrieval by microwave exposure in citrate buffer (pH 6.0), followed by incubation with rabbit anti-53BP1 (diluted 1:1000; Catalog No. IHC00001 Bethyl Laboratories, Montgomery, TX, USA) and monoclonal mouse anti-Ki-67 (MIB-1; 1:50; Catalog No. F7268; Dako, Jena, The Netherlands) antibodies. The specimens were subsequently incubated with the Alexa Fluor 488-conjugated goat anti-rabbit antibody (catalog no. A-11012; Molecular Probes Inc., Eugene, OR, USA) and Alexa Fluor 594 conjugated goat anti-mouse (Catalog No. A-11031; Molecular Probes Inc.) antibody prior to DAPI-I staining (VECTOR Laboratories, Burlingame, CA, USA). Lesions were analyzed and photographed at a magnification of 400× using a high-standard all-in-one fluorescence microscope (Biorevo BZ-9000; Keyence, Japan). We selected a field of observation in the superficial lesion, which was less than 100 µm from the surface. This depth was equivalent to the conditions encountered with biopsy specimens. During the evaluation of 53BP1 expression and its association with lymph and venous invasion in SCC, additional lesions at the front of the invasive tumor were photographed, and three areas per lesion were imaged (three areas per 125 lesions: a total of 375 areas).

### 2.4. Quantification of 53BP1 Nuclear Expression Pattern

The classification of 53BP1 nuclear expression was as follows: stable (no 53BP1 and/or one or two discrete nuclear foci measuring less than 1.0 µm), unstable (including at least one of the following features: three or more discrete nuclear foci [*n* ≥ 3], large nuclear foci [LF] exceeding 1.0 µm), and colocalization of 53BP1 and Ki67 (53BP1/Ki67) (Figure 2) [25,29]. The percentage of 53BP1 expression was also assessed.

### 2.5. Statistical Analysis

The Jonckheere–Terpstra test was conducted to examine the stepwise increase in 53BP1 and Ki67 expressions across non-tumor, LD, HD, and SCC lesions. Differences in 53BP1 expression between different tumor thicknesses, as well as the presence or absence of lymphovascular invasion, were assessed using the Wilcoxon signed-rank test. Logistic regression was employed for each factor that contributed to the variations observed between the non-tumor epithelium, LD, HD, and SCC lesions. The Youden index was used to identify the optimal cutoff points on the ROC curve. JMP Pro 16.0.0 software (SAS Institute, Inc., Cary, NC, USA) was used for statistical analysis. Statistical significance was set at *p* < 0.05.

## 3. Results

### 3.1. Patient Characteristics

Patient background characteristics are shown in Table 1. The T classifications of ESD/surgical cases, as per the TNM Classification of Malignant Tumors 8th edition [31], were as follows: Tis 5/1, T1 14/5, T2 3/4, T3 2/1, and T4 0/4 cases. The clinical features are shown in Table 2 [32]. Lymphatic and vascular invasion were observed in 23% and 20% of cases, respectively, and both were observed in 13% of the cases. The follow-up rate of all the cases employed in this study was 56%. In total, 92% of the patients were male, and the mean age was 66 ± 7.62 (SD). Multifocal and metachronous cancers were observed in 15% and 26% of the patients, respectively. A history of pharyngeal or esophageal cancer was present in 67% of the patients. The mean corpuscular volume (MCV) was 91.8 ± 12.0 (±SD, a typical adult level; 80–100 fl) and the median of γ-Glutamyl Transpeptidase (γGTP) was 27 (95%CI, 19–41 IU/l). Among all of the patients, the average alcohol intake was 42 ± 14.7 (g/day; mean ± SD) and the smoking rate was 44 ± 32.1 (pack years: number of cigarette packs per day × years; mean ± SD) (Table 1).

### 3.2. Progressive Increments in 53BP1 Nuclear Foci at the Tumor Surface during Hypopharyngeal Carcinogenesis

The expression pattern of 53BP1 foci was initially evaluated at the surface of non-tumor, LD, HD, and SCC lesions. The extent of total unstable 53BP1 as well as large foci exhibited a stepwise increase during progression from non-tumorous to SCC (Figure 3a,b, Appendix A). Co-expressions of Ki67 and 53BP1 at the tumor surface also increased as atypia progressed from LD to SCC (Figure 3a,b, Appendix A). Patient characteristics, including age, sex, cancer stage, metachronous recurrence, alcohol consumption, and smoking history, were not significantly associated with the type of 53BP1 nuclear foci in the squamous epithelium.

Next, we examined whether 53BP1 expression patterns could be used to distinguish HD and SCC, which are indications for ESD. When the receiver operating characteristic (ROC) curve was drawn to distinguish the indication for endoscopic treatment (non-tumor/LD vs. HD/SCC), the area under the curve (AUC) was 0.858, and the cutoff value was 3.251 for the unstable type, with 86.5% sensitivity and 78.9% specificity. The LF component of the unstable 53BP1 type exhibited the highest sensitivity among the examined parameters (86.8%) (Table 3). Based on these results, we performed multivariate analysis to distinguish non-tumor/LD from HD/SCC. Unstable 53BP1 (*n* ≥ 3 + LF) expression served as an independent factor that separated the two groups (odds ratio 1.561, 95% CI: 1.276–1.909; *p* < 0.0001) (Table 4).

### 3.3. Association of 53BP1 Nuclear Foci with Tumor Stage and Depth

We subsequently assessed the expression of 53BP1 and Ki67 in relation to the T classification according to the 8th edition of the TNM Classification of Malignant Tumors. Interestingly, the T classification revealed no significant differences between the different stages at the superficial and invasive front areas of the tumors (Figure 4). As a tumor thickness of ≥1000 µm is associated with an increased risk of lymph node metastasis [14,15,16], we assessed whether 53BP1 nuclear expression exhibited a different expression pattern in tumors with thicknesses of ≥1000 µm and <1000 µm. 53BP1 foci expression was observed at higher frequencies in SCC lesions with a thickness ≥1000 µm than in those with a thickness <1000 µm (Figure 5a). 53BP1/Ki67 colocalization foci exhibited statistically significant increases at the surface of SCC lesions with a thickness ≥1000 µm compared with those with a thickness <1000 µm (*p* < 0.05; Figure 5b). In contrast, unstable and large foci were not significantly different between the two groups. When the ROC curve was plotted for 53BP1/Ki67 to distinguish between tumor thicknesses ≥1000 µm and <1000 µm, the AUC was 0.61957, with a cut-off value of 0.969771.

### 3.4. The Presence of 53BP1 Nuclear Foci at the Invasive Front Is Inversely Correlated with Lymphovascular Invasion in HPSCC

Next, we examined whether 53BP1 expression patterns differ between the invasive front of HPSCC and the tumor surface. Overall, although Ki67 expression was higher in deeper lesions than in superficial lesions, the extent of 53BP1 nuclear foci formation did not differ significantly between the superficial and invasive front areas of HPSCC lesions (Figure 6).

Next, we examined whether the 53BP1 expression pattern was associated with lymphovascular invasion. Interestingly, on the surface of the tumor, 53BP1 nuclear foci patterns were not significantly different between the areas with the presence and absence of lymphovascular invasion (Figure 7).

In contrast, at the invasive front of the tumors, the numbers of unstable foci, large foci, and the colocalization of 53BP1/Ki67 were all significantly reduced (*p* < 0.05) in cases of lymph vessel invasion (Figure 8). Similarly, unstable and large foci significantly decreased in cases of venous invasion in the invasive front of the tumors (Figure 8).

## 4. Discussion

This study describes the progressive increments in 53BP1 foci expression during HPSCC carcinogenesis. The findings reveal a higher presence of 53BP1 and/or Ki67 at the surface of tumors exceeding 1000 µm in thickness, along with a decrease in 53BP1 nuclear foci formation in the invasive front of cancers with lymphovascular invasion. 

The expression of Ki67, a well-known proliferation marker that correlates with tumor malignancy and prognosis [33], did not differ significantly with changes in dysplasia status, tumor depth, or vascular invasion. However, the nuclear foci of 53BP1 exhibited a stepwise increase during carcinogenesis. Our results strengthen the findings of our previous experiments on the cervix, esophagus, and oral mucosa based on immunofluorescence analysis using formalin-fixed tissues. The frequent appearance of 53BP1 nuclear foci in human carcinogenesis suggests that the endogenous activation of DNA damage pathways in cancer cells is a hallmark of genomic instability [9,25,26]. For example, Matsuda demonstrated that the unstable expression of 53BP1 foci is correlated with chromosomal abnormalities in chromosomes 3, 6, 9p21, and 17 using fluorescence in situ hybridization (FISH) analysis [30]. In our study, the rate of unstable 53BP1 foci was the best predictor for endoscopic intervention. Thus, abnormal 53BP1 foci may predict the indication for early intervention in modern endoscopic treatment [11]. The progressive increase in 53BP1 foci expression during HPSCC carcinogenesis is in line with other studies examining squamous cell epithelia in the oral, cervical, and esophageal regions [9,25,26]. A 2021 report on oral squamous epithelial lesions by Imaizumi et al. [9] also suggested that 53BP1 expression increased with increasing histological malignancy. They reported a significant difference in 53BP1 expression between low-grade and high-grade dysplasia. Similarly, our study indicated a significant difference between low-grade and high-grade dysplasia (LD and HD) in unstable type (*n* > 3 + LF). This study differs from Imaizumi et al.’s report in that we incorporated a clinical perspective on the indication of treatment and analyzed tumor invasion using 53BP1 expression. In addition, some distinct findings regarding 53BP1 expression in HPSCC compared to previous studies were noted. For example, previous studies have proposed that large 53BP1 foci, which exceed 1 μm in diameter, are the best predictive factor for distinguishing HD [25,26], whereas this tendency was relatively modest in HPSCC. Although the underlying mechanisms responsible for these differences are unclear, our findings highlight the importance of investigating organ-specific nuclear 53BP1 foci patterns to aid diagnosis and prognosis predictions in malignancies. Notably, an evaluation of 53BP1 and Ki67 expressions in the context of the T classification according to the TNM Classification of Malignant Tumors 8th edition [31] revealed no significant differences between the different stages. The T classification system in the hypopharynx categorizes tumors based on their size and invasion into neighboring structures in the head and neck [31]. However, the T classification system does not consider the tumor depth within the hypopharynx, which may be the reason behind these results.

As for an assessment of tumor depth, The World Health Organization (WHO) 2017 classification of head and neck tumors does not incorporate depth of invasion (DOI) in the T classification for pharyngeal cancer as it does for oral cavity tumors [7]. However, since the release of the WHO 2017 classification of head and neck tumors, the endoscopic resection of superficial pharyngeal cancer has become more prevalent. Due to the absence of the muscularis mucosae in the pharynx, distinguishing between the intraepithelial (EP) and subepithelial (SEP) layers remains challenging. Consequently, the WHO classification alone is insufficient to accurately assess the invasion of superficial pharyngeal cancer in clinical practice. In this study, we included both endoscopically and surgically resected lesions and used tumor thickness, a metric commonly used in Japan [14,15,16,28], to evaluate the invasion of superficial pharyngeal cancer.

The high incidence of lymphatic and systemic spread could be responsible for the poor prognosis of HPSCC. The characteristics of the deep invasive front of SCCs often differ from those of the tumor surface and serve as a better prognostic indicator [33,34]. A tumor depth ≥1000 µm is a risk marker for lymphovascular invasion, a condition that potentially leads to lymph node and distant metastases [16]. However, currently, no methods for distinguishing the depth of HPSCC have been established. Our results suggest that 53BP1/Ki67 colocalization at the tumor surface is significantly higher in tumors with a depth ≥1000 µm than in those with a depth <1000 µm at the tumor surface. Given that the DDR pathway in healthy conditions is activated after cell cycle arrest by the p53 pathway, 53BP1/Ki67 colocalization is indicative of impaired DNA damage response machinery [26]. Our results indicate that 53BP1/Ki67 expression could be a factor in predicting the risk of metastasis in biopsy specimens. Prospective studies employing biopsy specimens are warranted to validate these findings.

Although 53BP1 serves as a surrogate marker of carcinogenesis, reduced 53BP1 foci expression has been reported upon the progression of tumors to more aggressive phenotypes [25]. Cells at the invasive tumor front cannot respond to DNA damage, indicating a possibility of augmented genomic instability [35]. In our study, 53BP1 expression was significantly reduced at the lymphovascular invasion sites of tumors. Interestingly, these differences were not observed at the tumor surface. In low-grade and high-grade dysplasia, DNA repair mechanisms remain active, limiting aggressive cancer proliferation. However, as malignancy progresses, DNA damage response (DDR) is impaired [36]. This mechanism explains the results obtained in this study. Our observation is in line with previous reports indicating a loss of reactivity to the DNA damage response during metastatic transformation [25].

53BP1 is known to negatively regulate epithelial–mesenchymal transition (EMT) [37], which may partially explain lymphovascular invasion consequent to the loss of 53BP1 expression during cancer invasion. However, further functional studies are required to determine the role of 53BP1 in the EMT of HPSCC. 

The limitations of this study include its small sample size and its retrospective nature. Among the 39 patients included in this study, 22 (56%) continued to be followed up at our hospital. Among these 22 cases, the 5-year overall survival rate was 88% and the 5-year cause-specific survival rate was 96%. Although this study experienced a significant dropout rate, the accumulation of additional cases and prospective follow-up may allow for a more accurate elucidation of the relationship between survival rates and pathological characteristics. A prospective analysis with a larger sample size has the potential to estimate the risk of metastasis in patients with HPSCC. Furthermore, it may be feasible to predict tumor depth and infiltration through biopsy alone, which could also help determine the necessity for additional treatments, such as chemoradiotherapy (CRT) or prophylactic lymph node dissection, following an endoscopic resection. Lastly, additional studies are required to pinpoint how the DNA damage response induces the progression of malignancies in HPSCC, which include the analysis of chromosomal rearrangement by FISH and DNA copy number aberration by microarray, as well as epigenetic changes in DNA methylation by employing next-generation sequencing.

## 5. Conclusions

In conclusion, we demonstrated that, during HPSCC carcinogenesis, the presence of 53BP1 nuclear foci increased with progression from non-tumorous to LD, HD, and SCC, indicating a stepwise increase in genomic instabilities. In addition, the colocalization of 53BP1 foci and Ki67 was increased in tumors with a depth of ≥1000 µm, compared to those <1000 µm in depth at the tumor surface, implying that a DNA damage response occurring at abnormal timing could be detected in biopsy specimens to predict the depth of the tumor. Furthermore, 53BP1 expression was diminished in the invasive front of the tumor with lymphovascular invasion, indicating a loss of DNA damage response in these conditions. Monitoring aberrant 53BP1, particularly in conjunction with Ki67, may enhance the predictive accuracy for lymph node metastasis and vascular invasion, ultimately aiding the prognosis and management of patients with HPSCC. 

## Figures and Tables

**Figure 1 cancers-16-02987-f001:**
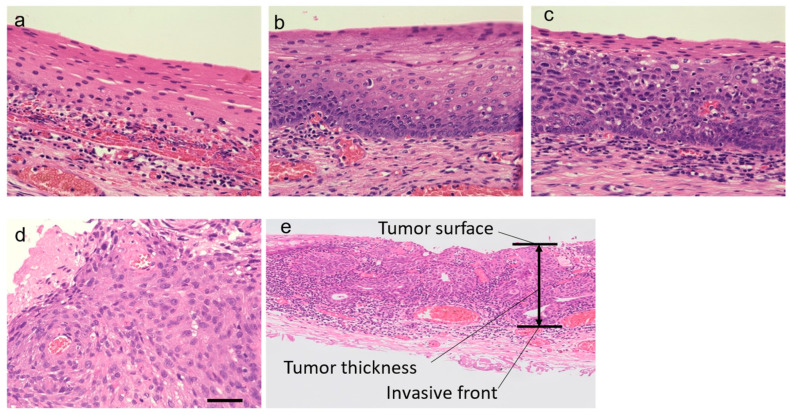
Representative Hematoxylin–Eosin staining of squamous epithelium employed in our study (original magnification: ×40): (**a**) non-tumor, (**b**) low-grade dysplasia, (**c**) high-grade dysplasia, and (**d**) squamous cell carcinoma. Scale bar = 50 µm. (**e**) Tumor thickness indicates the distance from the tumor surface to its deepest point.

**Figure 2 cancers-16-02987-f002:**
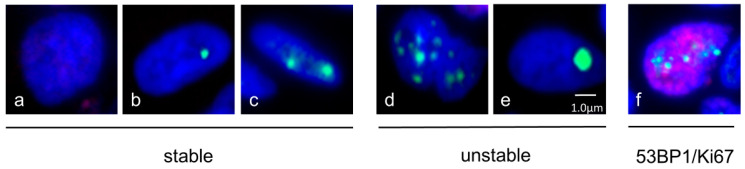
Representative images of nuclear expression patterns of 53BP1. Stable (**a**) zero, (**b**) one, or (**c**) two foci; (**d**) unstable; (≥3 nuclear foci); (**e**) large foci (≥1 μm in diameter); (**f**) colocalization of 53BP1 foci and Ki67. 53BP1 (green), Ki67 (red), and DAPI (blue).

**Figure 3 cancers-16-02987-f003:**
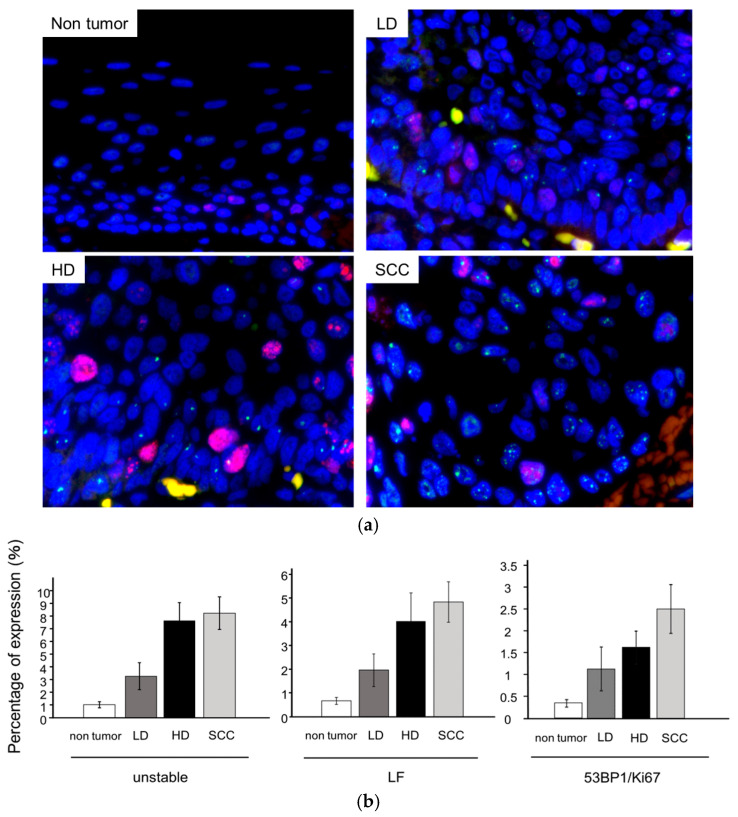
Expression patterns of 53BP1 nuclear foci during the progression of squamous neoplasms in the hypopharynx. (**a**) Representative image of double-label immunofluorescence staining of 53BP1 and Ki-67 in the hypopharynx. LD, low-grade dysplasia; HD: high-grade dysplasia; SCC: squamous cell carcinoma. 53BP1 (green), Ki67 (red), and DAPI (blue). Original magnification: ×400. (**b**) Quantification of stepwise increase in the following nuclear expression patterns: unstable expression, large foci (LF), and colocalization of 53BP1/Ki67 during carcinogenesis. *p* < 0.0001, Jonckheere–Terpstra test. Data were calculated as a percentage (%) of the total number of epithelial cells and are presented as mean ± standard error of the mean (SEM). LD, low-grade dysplasia; HD: high-grade dysplasia; SCC: squamous cell carcinoma. LF: large nuclear foci, 53BP1/Ki67: colocalization of 53BP1 and Ki67.

**Figure 4 cancers-16-02987-f004:**
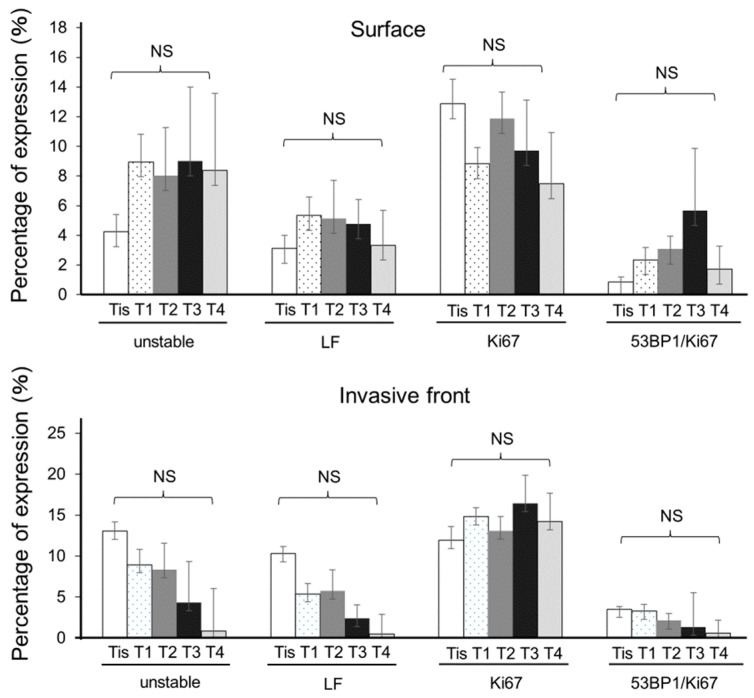
Association between 53BP1 expression pattern and T factor of TNM classification. LF: large nuclear foci; 53BP1/Ki67: colocalization of 53BP1 and Ki67. Data are expressed as mean ± SEM; NS, not significant.

**Figure 5 cancers-16-02987-f005:**
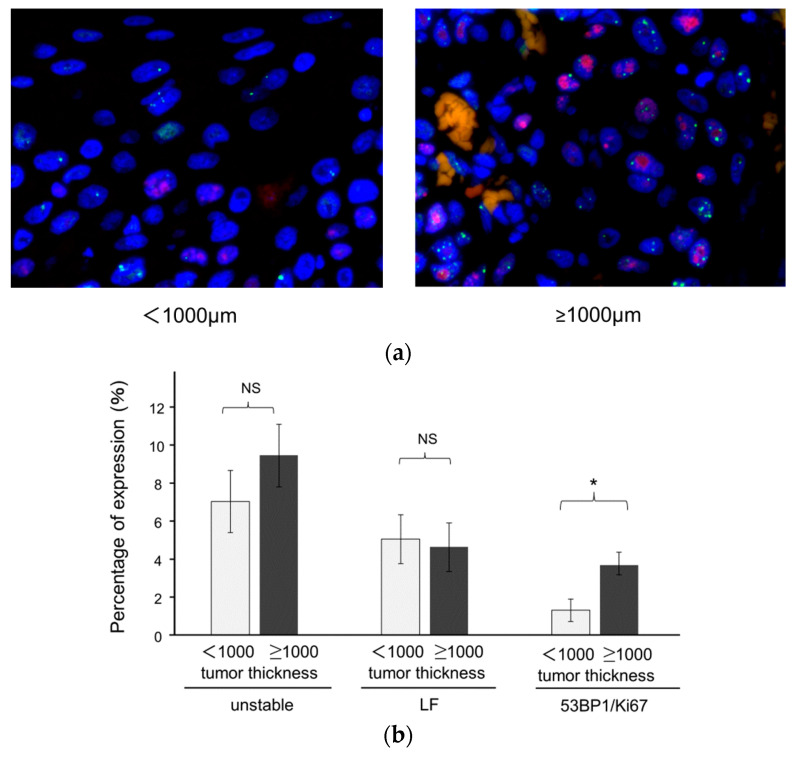
Expression pattern of 53BP1 nuclear foci according to tumor depth. (**a**) Representative images of 53BP1 (green) and Ki67 (red) staining in tumors with a thickness <1000 µm and ≥1000 µm at the surface of squamous cell carcinoma (SCC) visualized by 53BP1 (green) and Ki67 (red) staining with nuclei stained with DAPI (blue). Original magnification: ×400. (**b**) Quantification of unstable, large foci (LF), and colocalization of 53BP1 and Ki67 (53BP1/Ki67) at the surface of HPSCC with a tumor thickness <1000 µm and ≥1000 µm. Data are expressed as mean ± SEM; NS, not significant; * *p* < 0.05.

**Figure 6 cancers-16-02987-f006:**
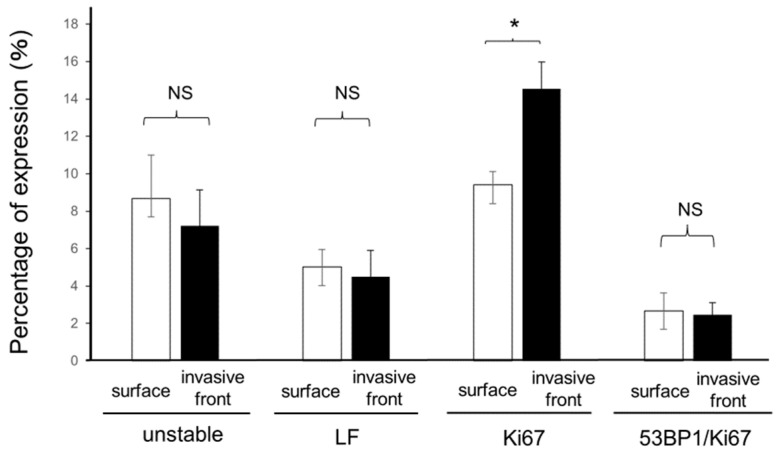
Comparison of 53BP1 expression patterns between the tumor surface and invasive front. LF: large nuclear foci; 53BP1/Ki67: colocalization of 53BP1 and Ki67. Data were calculated as a percentage (%) of the total number of epithelial cells and are presented as mean ± SEM; NS, not significant, * *p* < 0.05.

**Figure 7 cancers-16-02987-f007:**
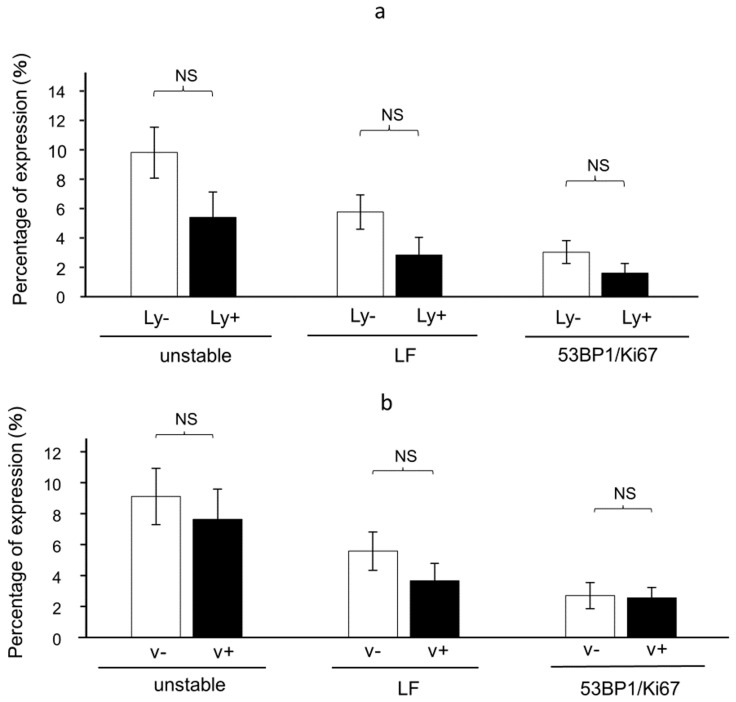
Expression pattern of 53BP1 nuclear foci according to lymphatic invasion. (**a**) Quantification of 53BP1 expression at the tumor surface in the presence or absence of lymphatic invasion (Ly+/Ly−). Data are expressed as mean ± SEM. (**b**) Quantification of 53BP1 and Ki67 at the tumor surface in the presence or absence of vascular invasion (v+/v−). LF: large nuclear foci; 53BP1/Ki67: colocalization of 53BP1 and Ki67. Data are expressed as mean ± SEM; NS, not significant.

**Figure 8 cancers-16-02987-f008:**
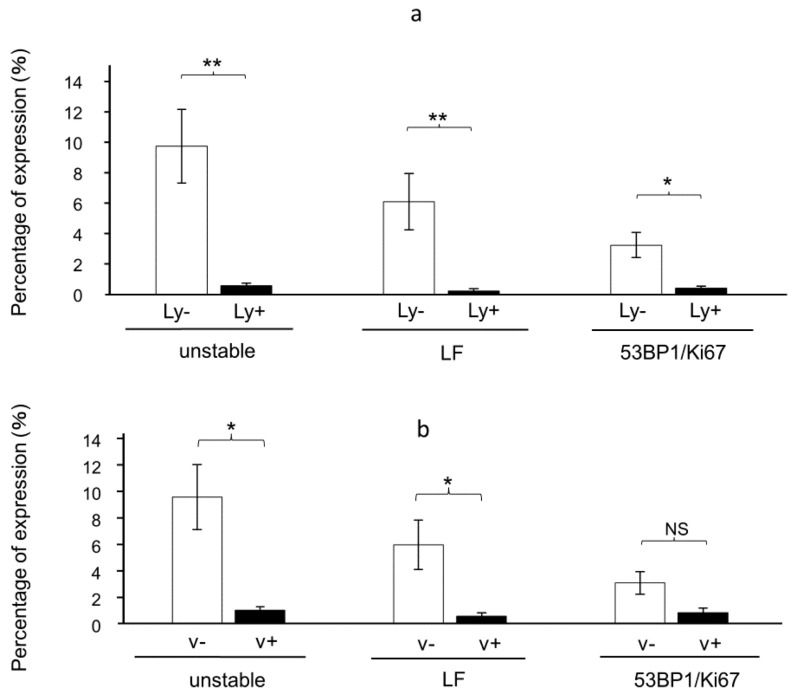
Expression pattern of 53BP1 nuclear foci according to venous invasion. (**a**) Quantification of 53BP1 expression patterns at the tumor invasion area, in the presence or absence of lymphatic invasion (Ly+/Ly−). LF: large nuclear foci; 53BP1/Ki67: colocalization of 53BP1 and Ki67. Data are expressed as mean ± SEM. (**b**) Quantification of 53BP1 and Ki67 at the tumor invasion area in the presence or absence of vascular invasion (v+/v−). Data are expressed as mean ± SEM; NS, not significant; * *p* < 0.05, ** *p* < 0.001.

**Table 1 cancers-16-02987-t001:** Patient characteristics.

	*n* = 39
Sex	
Male, *n* (%)	36 (92%)
Female, *n* (%)	3 (8%)
Age, mean ± SD	66 ± 7.62
Multifocal, *n* (%)	6 (15%)
Metachronous, *n* (%)	10 (26%)
History of pharyngeal/esophagus cancer, *n* (%)	26 (67%)
MCV, mean ± SD	91.8 ± 12.0
γGTP, median [95%CI]	27 [19–41]
Alcohol, g/day, median [95%CI]	50 [62–70]
Alcohol, g/year, mean ± SD	42 ± 14.7
Smoking pack years, mean ± SD	44 ± 32.1

SD: standard deviation, MCV: mean corpuscular volume, γGTP: γ-Glutamyl transpeptidase.

**Table 2 cancers-16-02987-t002:** Clinical features.

cTNM, *n* (%)	
cTis/1/2/3/4	6 (15)/19 (48)/7 (17)/3 (7)/4 (10)
cN 0/1/2b/2c/3b	32 (82)/3 (7)/1 (2)/2 (5)/1 (2)
cM 0/1	38 (97)/1 (2)
cStage 0/I/II/III/IVA/IVB/IVC	6 (15)/17 (43)/4 (10)/5 (12)/5 (12)/1 (2)/1 (2)
Lympho-vascular invasion	
Ly+, *n* (%)	9 (23)
v+, *n* (%)	8 (20)
Ly and v, +, *n* (%)	5 (13)
Follow up rate, *n* (%)	22 (56)

cTNM: clinical TNM classification, Ly+: lymphatic invasion, v+: vascular invasion.

**Table 3 cancers-16-02987-t003:** Logistic analysis of the factors related to the indication of the treatment.

Logistic Analysis							
Factors	OR	95%CI	*p*	Sensitivity	Specificity	Cutoff	AUC
Unstable expression	1.495	1.26–1.78	<0.0001	0.789	0.865	3.251	0.858
*n* ≥ 3	2.13	1.49–3.05	<0.0001	0.816	0.769	0.787	0.837
LF	1.816	1.37–2.40	<0.0001	0.868	0.769	1.46	0.834
Ki67	1.084	0.99–1.19	0.0845	0.447	0.827	10.784	0.629
53BP1/Ki67	1.864	1.25–2.79	0.0025	0.605	0.808	0.96	0.747

OR: odds ratio, LF: large nuclear foci, and 53BP1/Ki67: colocalization of 53BP1 and Ki67. AUC: area under the curve.

**Table 4 cancers-16-02987-t004:** Multivariate analysis of the factors contributing to treatment.

	OR	95%CI		*p*-Value
Unstable 53BP1 (*n* ≥ 3 + LF) at surface	1.56107	1.276307	1.909368	<0.0001
Sex (male/female)	0.69938	0.093741	5.217704	0.7273
Age	1.002435	0.922729	1.089025	0.9541
Multifocal	1.430849	0.273848	7.476156	0.6711
Metachronous	2.117932	0.525765	8.531632	0.2911
History of pharyngeal/esophagus cancer	0.371583	0.098147	1.406806	0.1450
Alcohol, g/day	0.996389	0.988839	1.003997	0.3513
Smoking pack years	1.001769	0.984427	1.019417	0.8428

OR: odds ratio.

## Data Availability

The data that support the findings of this study are available from the corresponding author, Y.A., upon reasonable request.

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
