# Peer review of "Significance of P53-Binding Protein 1 as a Novel Molecular Histological Marker for Hypopharyngeal Squamous Neoplasms"

_cancers, 2024, doi:10.3390/cancers16172987_

Round 1

Reviewer 1 Report

Comments and Suggestions for Authors

In this study, authors evaluated 53BP1 expression pattern as a biomarker predicting progression of a hypopharyngeal lesion to a more aggressive form in 39 patients. The introduction, methods and discussion sections were well-written and comprehensive. The result section could benefit from the following:

Number of cases in each lesion category, i.e LD, HD, and SCC including in situ and invasive. 

Number of cases with each expression pattern in each lesion category. 

Author Response

Response to Reviewer 1 Comments

Thank you very much for taking the time to review our manuscript and for offering valuable suggestions. Please find the detailed responses below and the corresponding revisions highlighted in the re-submitted files.

Point-by-point response to Comments and Suggestions for Authors

Comments 1: The result section could benefit from the following: Number of cases in each lesion category, i.e LD, HD, and SCC including in situ and invasive. 

Response 1: Thank you very much for your suggestion.

We have added the relevant information on the cases, as shown below, on page 3 line 129 to 131.

Lesions were classified as non-neoplastic from outside the tumor margin (38 sites/38 cases), LD (14 sites/14 cases), HD (8 sites/8 cases), or SCC (surface lesion 30 sites/30 cases, invasive front, 35 sites/35 cases), with a total of 125 sites (online resource 1).

The number of cases is also shown in the Results, section 3.1. Patient characteristics, on page 5 line 179.

Comments 2: The result section could benefit from the following: Number of cases with each expression pattern in each lesion category.

Response 2: Thank you very much for your helpful comment.

We agree, and have generated the table shown below, on page 3 line 131, and page 6 line 200, 201 as online resource 1.

I hope we have adequately addressed your comments.

Online resource 1

Lesions

sites (cases)

Counted

nuclei

Stable 53BP1

Unstable 53BP1

Ki67

Positive nuclei

53BP1/Ki67

colocalization

n=0

n=1,2

n>3

LF

Non tumor

38 (38)

11646

11011

(94.24ï¼…)

(38)

348

(3.04ï¼…)

(38)

828

(6.92ï¼…)

(15)

45

(0.34ï¼…)

(23)

828

(6.92ï¼…)

(35)

45

(0.34ï¼…)

(16)

LD

14 (14)

6073

5343

(86.73ï¼…)

(14)

331

(5.87ï¼…)

(14)

576

(9.77ï¼…)

(10)

63

(1.13ï¼…)

(9)

576

(9.77ï¼…)

(14)

63

(1.13ï¼…)

  (10)

HD

8 (8)

3376

3099

(81.61ï¼…)

(8)

369

(10.04ï¼…)

(8)

314

(8.58ï¼…)

(7)

59

(1.61ï¼…)

(8)

314

(8.58ï¼…)

(8)

59

(1.61ï¼…)

(7)

SCC

Surface

30 (30)

9727

7914

(83.44ï¼…)

(30)

838

(8.23ï¼…)

(30)

912

(9.61ï¼…)

(27)

232

(1.32ï¼…)

(27)

912

(9.61ï¼…)

(30)

232

(1.32ï¼…)

(24)

SCC

invasive front

35 (35)

15472

12825

(76.81ï¼…)

(35)

1215

(10.35ï¼…)

(35)

2055

(12.75ï¼…)

(28)

407

(3.26ï¼…)

(29)

2055

(12.75ï¼…)

(35)

407

(3.26ï¼…)

(30)

p<0.0001

p<0.0001

LD: low-grade dysplasia; HD: High-grade dysplasia; SCC: Squamous cell carcinoma. 53BP1/Ki67: 53BP1 and Ki67.

Reviewer 2 Report

Comments and Suggestions for Authors

The work done by Hiroko et al. demonstrates that the DNA damage response protein p53-binding protein 1 (53BP1) exhibits abnormal foci during carcinogenesis in various organs, including hypopharyngeal squamous cell carcinoma (HPSCC). Their findings suggest that 53BP1 could be useful for the pathological diagnosis and prognosis of hypopharyngeal cancer.

To further enhance the paper and make it more engaging for readers, I have a few comments and suggestions that I believe would add valuable context and depth to your discussion:

·        The authors have demonstrated interesting findings regarding 53BP1 expression in HPSCC. Can you comment on how these results compare to other head and neck cancers?

·        What molecular mechanisms might explain the reduction in 53BP1 expression at lymphovascular invasion sites.

·        Have you considered examining other DNA damage response proteins or cell cycle regulators?

·        Your study revealed differences in 53BP1 expression between the tumor surface and invasive front. How might this impact biopsy practices, considering that biopsies typically sample the tumor surface.

Lastly, the authors mention the limitations of sample size and retrospective nature. Can they discuss potential future directions for prospective studies that could further validate and extend these findings.

Comments on the Quality of English Language

Minor editing of English language required

Author Response

Response to Reviewer 2 Comments

Thank you very much for taking the time to review our manuscript and for offering valuable suggestions. Please find the detailed responses below and the corresponding revisions highlighted in the re-submitted files.

Point-by-point response to Comments and Suggestions for Authors

Comments 1: The authors have demonstrated interesting findings regarding 53BP1 expression in HPSCC. Can you comment on how these results compare to other head and neck cancers?

Response 1: Thank you very much for your important question.

A 2021 report on oral squamous epithelial lesions by Imaizumi et al. [1] also revealed that 53BP1 expression increased with increasing histological malignancy. They reported a significant difference in 53BP1 expression between low-grade and high-grade dysplasia. Similarly, our study showed a significant difference between low-grade and high-grade dysplasia (LD and HD) in unstable type (n > 3 + LF).

However, this study differs from Imaizumi et al.'s report in that we incorporated a clinical perspective on the indication of treatment and analyzed tumor invasion using 53BP1 expression.

Additional explanation has been added to the Discussion (page 13 from line 313 to line 320).

Comments 2: What molecular mechanisms might explain the reduction in 53BP1 expression at lymphovascular invasion sites.

Response 2: Thank you for pointing this out.

In low-grade and high-grade dysplasia, DNA repair mechanisms remain active, limiting aggressive cancer proliferation. However, as malignancy progresses, the DNA damage response (DDR) is impaired [2]. This mechanism explains the results in this study.

Additional explanation has been added to the Discussion (page 14 from line 362 to line 365). 

Comments 3:  Have you considered examining other DNA damage response proteins or cell cycle regulators?

Response 3: Thank you for your important question.

53BP1 is a key regulator of non-homologous end-joining (NHEJ) repair. During carcinogenesis, we consider tumor protein 53-binding protein 1 (53BP1) as a nuclear protein that accumulates and forms abnormal foci in response to double-strand DNA breaks, due to lack of downstream regulation. Previous studies have suggested that 53BP1 is expressed along with other DNA damage response molecules, including gamma histone 2AX (γH2AX) and ataxia telangiectasia mutated (ATM), during carcinogenesis [3–5]. 

The above text has been included on page 2, line 81 to 84. 

Comments 4:  Your study revealed differences in 53BP1 expression between the tumor surface and invasive front. How might this impact biopsy practices, considering that biopsies typically sample the tumor surface.

Response 4: Thank you for pointing this out.

Currently, there is no consensus on predicting the depth of invasion using imaging techniques alone, such as narrow banding image (NBI) of endoscopy, CT, or MRI. Although future prospective studies are required, biopsy samples may offer a reliable method for predicting patient prognosis.

Comments 5: Lastly, the authors mention the limitations of sample size and retrospective nature. Can they discuss potential future directions for prospective studies that could further validate and extend these findings.

Response 5: Thank you for your comment.

In the future, it may be feasible to predict tumor depth and infiltration through biopsy alone, which could also help determine the necessity for additional treatments, such as chemoradiotherapy (CRT) or prophylactic lymph node dissection, following endoscopic resection.

Additional explanation has been added to the Discussion (page 14 from line 379 to line 382).

References

  1. Imaizumi, T.; Matsuda, K.; Tanaka, K.; Kondo, H.; Ueki, N.; Kurohama, H.; Otsubo, C.; Matsuoka, Y.; Akazawa, Y.; Miura, S.; et al. Detection of endogenous DNA double-strand breaks in oral squamous epithelial lesions by P53-binding Protein 1. Anticancer Res. 2021, 41, 4771–4779. DOI:10.21873/anticanres.15292.
  2. O’Connor, M.J. Targeting the DNA damage response in cancer. Mol. Cell 2015, 60, 547–560. DOI:10.1016/j.molcel.2015.10.040.
  3. Bork, P.; Hofmann, K.; Bucher, P.; Neuwald, A.F.; Altschul, S.F.; Koonin, E.V. A superfamily of conserved domains in DNA damage-responsive cell cycle checkpoint proteins. FASEB J. 1997, 11, 68–76. DOI:10.1096/fasebj.11.1.9034168.
  4. Ward, I.M.; Minn, K.; Jorda, K.G.; Chen, J. Accumulation of checkpoint protein 53BP1 at DNA breaks involves its binding to phosphorylated histone H2AX. J. Biol. Chem. 2003, 278, 19579–19582. DOI:10.1074/jbc.C300117200.
  5. Schultz, L.B.; Chehab, N.H.; Malikzay, A.; Halazonetis, T.D. p53 binding protein 1 (53BP1) is an early participant in the cellular response to DNA double-strand breaks. J. Cell Biol. 2000, 151, 1381–1390. DOI:10.1083/jcb.151.7.1381.

Reviewer 3 Report

Comments and Suggestions for Authors

The study explores the potential of 53BP1 on hypopharyngeal squamous cell carcinomas (HPSCC). The authors found that 53BP1 expression increases with the progression from normal tissue, dysplastic lesions, to HPSCC, suggesting that 53BP1 may predict progression of hypopharyngeal tissues during carcinogenesis. The authors also showed that the expression of 53BP1 in the surface of tumors exceeding 1000 μm in thickness, along with a decrease in 53BP1 nuclear foci formation in the invasive front of the cancer, may be related to lymphovascular invasion. It was a pleasure to read this interesting and well-conducted study, but I have some comments in order to improve the manuscript.

1. Figure 1D. The suggestion is to display the representative HPSCC case in lower magnification, allowing a view of the tumor invasion into the adjacent connective tissue. Indeed, as tumor thickness influences the pattern of 53BP1 expression, the readers of the study would benefit if the authors could illustrate both superficial and deep (invasive front) lesions, adding the measurement. Furthermore, the authors never explain how tumor thickness was determined.

2. Depth of invasion and tumor thickness are commonly used interchangeably, and the main explanation for this fact is that head neck cancers, including HPSCC, present commonly as an ulceration, and the inclusion or not of few epithelial layers into measurement adds minimal depth. However, there are clear definitions to consider them different, and the World Health Organization (WHO) 2017 classification of head and neck tumors adopts depth of invasion. Why was tumor thickness used instead of depth of invasion? Make this clear in the discussion of the study.

3. Regarding quantification of the immunofluorescence, the authors should explain how the areas (3 for each site) were selected. How are they representative of the whole section? I would like to hear the opinion of the authors whether it would be better or not to normalize the quantification by number of nuclei instead of area of the microphotography.

4. T stage (or T category) is different from clinical stage. Please define TNM clinical stage and confirm that all patients were N0 and M0.

5. There were 6 cases classified as in situ carcinoma (Tis). In which group were these patients included? As high-grade dysplasia, since the treatment recommended is the same, or as invasive carcinoma? Why were they not put in an independent group?

6. The authors should describe the number of cases with lymphovascular invasion (LVI), and state how lymphatic and blood vessels were distinguished? Without specific markers, I strongly suggest that both should be considered together as one group (LVI). The result is not going to change. Moreover, please define LVI, or lymph vessel invasion and venous invasion, if that is the case. Only if tumor cells are within the vascular space, or both within and adjacent to the vessels.

7. Considering the small cohort (39 cases), the authors should mention whether clinicopathological association and survival analyses are possible to be performed.

Author Response

Response to Reviewer 3 Comments

Thank you very much for taking the time to review our manuscript and for offering valuable suggestions. Please find the detailed responses below and the corresponding revisions highlighted in the re-submitted files.

Point-by-point response to Comments and Suggestions for Authors

Comments 1: Figure 1D. The suggestion is to display the representative HPSCC case in lower magnification, allowing a view of the tumor invasion into the adjacent connective tissue. Indeed, as tumor thickness influences the pattern of 53BP1 expression, the readers of the study would benefit if the authors could illustrate both superficial and deep (invasive front) lesions, adding the measurement. Furthermore, the authors never explain how tumor thickness was determined.

Response 1: Thank you for your valuable comment.

We agree with your comment. Therefore, we have added the new pathological image as Figure 1e, illustrating both the superficial and invasive front, as well as tumor thickness.

This content had been added to Page 4 line 132.     

According to the General Rules for Clinical Studies on Head and Neck Cancer, 6th Edition, Revised Version (Japan Society for Head and Neck Cancer, December 2019, 86), tumor thickness can be defined as the distance from the tumor surface to its deepest point [1–4].

Additional explanation has been added to Histological evaluation (page 3 from line 121 to line 124).

Comments 2: Depth of invasion and tumor thickness are commonly used interchangeably, and the main explanation for this fact is that head neck cancers, including HPSCC, present commonly as an ulceration, and the inclusion or not of few epithelial layers into measurement adds minimal depth. However, there are clear definitions to consider them different, and the World Health Organization (WHO) 2017 classification of head and neck tumors adopts depth of invasion. Why was tumor thickness used instead of depth of invasion? Make this clear in the discussion of the study.

Response 2: Thank you very much for your important comment.

The World Health Organization (WHO) 2017 classification of head and neck tumors does not incorporate depth of invasion (DOI) in the T classification for pharyngeal cancer as it does for oral cavity tumors. However, since the release of the WHO 2017 classification of head and neck tumors, endoscopic resection of superficial pharyngeal cancer has become more prevalent. Due to the absence of the muscularis mucosae in the pharynx, distinguishing between the intraepithelial (EP) and subepithelial (SEP) layers remains challenging. Consequently, the WHO classification alone is insufficient to accurately assess the invasion of superficial pharyngeal cancer in clinical practice. In this study, we included both endoscopically and surgically resected lesions and used tumor thickness, a metric commonly used in Japan, to evaluate the invasion of superficial pharyngeal cancer [1–4].

Additional explanation has been added to the Discussion (page 13 from line 333 to line 343).

Comments 3: Regarding quantification of the immunofluorescence, the authors should explain how the areas (3 for each site) were selected. How are they representative of the whole section? I would like to hear the opinion of the authors whether it would be better or not to normalize the quantification by number of nuclei instead of area of the microphotography.

Response 3: Thank you very much for pointing this out.

In this study, we did quantify the number of nuclei within each region and calculated the percentage of the nuclei that were expressed by immunofluorescence staining of 53BP1 and/or Ki67. The representative areas for each assessment were randomly selected, as described previously [5–10].  I hope this adequately addresses your question.

This content has been added text to Histological evaluation, page 3 from line 125 to line 128.

Furthermore, we generated a table, which had been added to page 3 line 131, and page 6 line 200 and 201 as online resource 1.

Comments 4: T stage (or T category) is different from clinical stage. Please define TNM clinical stage and confirm that all patients were N0 and M0.

Response 4: Thank you very much for your valuable comment. We generated a table, as shown below, to present the clinical features of this study according to the UICC 8th edition. As surgically resected specimens were included in this study, not all cases were classified as N0M0.

Additional explanation has been added to the Results (page 5 from line 179 to line 182), and the table below had been added to page 6 line from 192 to 193.

UICC, n (%)

 cT is/1/2/3/4

6(15)/19(48)/7(17)/3(7)/4(10)

 cN 0/1/2b/2c/3b

32(82)/3(7)/1(2)/2(5)/1(2)

 cM 0/1

38(97)/1(2)

 cStage 0/I/II/III/IVA/IVB/IVC

6(15)/17(43)/4(10)/5(12)/5(12)/1(2)/1(2)

Lymphovascular invasion

 Ly +, n (%)

9 (23)

 v +, n (%)

8 (20)

Ly and v, +, n (%)

5(13)

Follow up rate, n (%)

22 (56)

Comments 5: There were 6 cases classified as in situ carcinoma (Tis). In which group were these patients included? As high-grade dysplasia, since the treatment recommended is the same, or as invasive carcinoma? Why were they not put in an independent group?

Response 5: Thank you very much for pointing this out.

In this study, specimens were classified based on the histological malignancy grade ( non-tumor, low-grade dysplasia, high-grade dysplasia, and squamous cell carcinoma) and tumor thickness, with a threshold of 1000 µm; thus, we did not to classify Tis separately. The maximum tumor thickness for Tis in this study was 550 µm, placing all Tis cases within the category of SCC tumors with a thickness of less than 1000 µm. In addition, the invasive front was only assessed in cases of stromal invasion.

Comments 6: The authors should describe the number of cases with lymphovascular invasion (LVI), and state how lymphatic and blood vessels were distinguished? Without specific markers, I strongly suggest that both should be considered together as one group (LVI). The result is not going to change. Moreover, please define LVI, or lymph vessel invasion and venous invasion, if that is the case. Only if tumor cells are within the vascular space, or both within and adjacent to the vessels.

Response 6: Thank you very much for pointing this out.

There were nine cases of lymphathic invasion and eight cases of vascular invasion. Among those cases, five exhibited lymphovascular invasion (please refer to the table of response 4). In this study, all cases were stained with D2-40 (lymphatic) and Elastica-van Gieson (EVG, blood vessels), in addition to routine HE staining, allowing for differentiation between lymphatic and vascular invasion. Invasion to the vessels was only confirmed when tumor cells were within the vascular space. 

Comments 7: Considering the small cohort (39 cases), the authors should mention whether clinicopathological association and survival analyses are possible to be performed.

Response 7: Thank you very much for your important comment.

Among the 39 patients included in this study, 22 (56%) continued to be followed up at our hospital. Among these 22 cases, the 5-year overall survival rate was 88% and the 5-year cause-specific survival rate was 96%. Although this study experienced a significant dropout rate, the accumulation of additional cases and prospective follow-up may allow for more accurate elucidation of the relationship between survival rates and pathological characteristics.

Additional explanation has been added to the Discussion (page 14 from line 373 to 378).

References

  1. Taniguchi, M.; Watanabe, A.; Tsujie, H.; Tomiyama, T.; Fujita, M.; Hosokawa, M.; Sasaki, S. Predictors of cervical lymph node involvement in patients with pharyngeal carcinoma undergoing endoscopic mucosal resection. Auris Nasus Larynx 2011, 38, 710–717. DOI:10.1016/j.anl.2011.01.001.
  2. Kinjo, Y.; Nonaka, S.; Oda, I.; Abe, S.; Suzuki, H.; Yoshinaga, S.; Maki, D.; Yoshimoto, S.; Taniguchi, H.; Saito, Y. The short-term and long-term outcomes of the endoscopic resection for the superficial pharyngeal squamous cell carcinoma. Endosc Int Open 2015, 3, E266–E273. DOI:10.1055/s-0034-1392094.
  3. Sasaki, T.; Kishimoto, S.; Kawabata, K.; Sato, Y.; Tsuchida, T. Risk factors for cervical lymph node metastasis in superficial head and neck squamous cell carcinoma. J Med Dent Sci 2015, 62, 19–24. DOI:10.11480/620103.
  4. Iritani, K.; Del Mundo, D.A.A.; Iwaki, S.; Masuda, K.; Kanzawa, M.; Furukawa, T.; Teshima, M.; Shinomiya, H.; Morimoto, K.; Otsuki, N.; et al. Prognostic factors after transoral resection of early hypopharyngeal cancer. Laryngoscope Investig Otolaryngol 2021, 6, 756–763. DOI:10.1002/lio2.611.
  5. Imaizumi, T.; Matsuda, K.; Tanaka, K.; Kondo, H.; Ueki, N.; Kurohama, H.; Otsubo, C.; Matsuoka, Y.; Akazawa, Y.; Miura, S.; et al. Detection of endogenous DNA double-strand breaks in oral squamous epithelial lesions by P53-binding Protein 1. Anticancer Res 2021, 41, 4771–4779. DOI:10.21873/anticanres.15292.
  6. Ueki, N.; Akazawa, Y.; Miura, S.; Matsuda, K.; Kurohama, H.; Imaizumi, T.; Kondo, H.; Nakashima, M. Significant association between 53 BP1 expression and grade of intraepithelial neoplasia of esophagus: alteration during esophageal carcinogenesis. Pathol Res Pract 2019, 215, 152601. DOI:10.1016/j.prp.2019.152601.
  7. Matsuda, K.; Miura, S.; Kurashige, T.; Suzuki, K.; Kondo, H.; Ihara, M.; Nakajima, H.; Masuzaki, H.; Nakashima, M. Significance of p53-binding protein 1 nuclear foci in uterine cervical lesions: endogenous DNA double strand breaks and genomic instability during carcinogenesis. Histopathology 2011, 59, 441–451. DOI:10.1111/j.1365-2559.2011.03963.x.
  8. Nakashima, M.; Suzuki, K.; Meirmanov, S.; Naruke, Y.; Matsuu-Matsuyama, M.; Shichijo, K.; Saenko, V.; Kondo, H.; Hayashi, T.; Ito, M.; et al. Foci formation of P53-binding protein 1 in thyroid tumors: activation of genomic instability during thyroid carcinogenesis. Int J Cancer 2008, 122, 1082–1088. DOI:10.1002/ijc.23223.
  9. Akazawa, Y.; Nakashima, R.; Matsuda, K.; Okamaoto, K.; Hirano, R.; Kawasaki, H.; Miuma, S.; Miyaaki, H.; Malhi, H.; Abiru, S.; et al. Detection of DNA damage response in nonalcoholic fatty liver disease via p53-binding protein 1 nuclear expression. Mod Pathol 2019, 32, 997–1007. DOI:10.1038/s41379-019-0218-8.
  10. Matsuda, K.; Kawasaki, T.; Akazawa, Y.; Hasegawa, Y.; Kondo, H.; Suzuki, K.; Iseki, M.; Nakashima, M. Expression pattern of p53-binding protein 1 as a new molecular indicator of genomic instability in bladder urothelial carcinoma. Sci Rep 2018, 8, 15477. DOI:10.1038/s41598-018-33761-9.

Round 2

Reviewer 3 Report

Comments and Suggestions for Authors

The authors have adequately answered and incorporated my suggestions in the revised manuscript. However, the answer to comment 6, which is appropriated, must be also incorporated into the manuscript. It is very relevant to the readers to have the knowledge that the samples were immunostained for D2-40 to identify the lymphatic vessels and Elastica-van Gieson staining was performed to characterize blood vessels. Moreover, the criterion adopted for invasion (tumor cells within the vascular space) should be clearly described.

Author Response

Response to Reviewer 3 Comments

Thank you very much for taking the time to review our manuscript and for offering valuable suggestions. Please find the detailed responses below and the corresponding revisions highlighted in the re-submitted files.

Point-by-point response to Comments and Suggestions for Authors

Comment:

The authors have adequately answered and incorporated my suggestions in the revised manuscript. However, the answer to comment 6, which is appropriated, must be also incorporated into the manuscript. It is very relevant to the readers to have the knowledge that the samples were immunostained for D2-40 to identify the lymphatic vessels and Elastica-van Gieson staining was performed to characterize blood vessels. Moreover, the criterion adopted for invasion (tumor cells within the vascular space) should be clearly described.

Response:

Thank you very much for your important comment.

The content below has been added to Histological evaluation, page 3, from line 121 to line 125

In this study, all cases were stained with D2-40 (lymphatic) and Elastica-van Gieson (EVG, blood vessels), in addition to routine Hematoxylin Eosin (HE) staining, allowing for differentiation between lymphatic and vascular invasion. Invasion to the vessels was only confirmed when tumor cells were within the vascular space.
